# Proposal for an Integrative Cognitive-Emotional Conception of ADHD

**DOI:** 10.3390/ijerph192215421

**Published:** 2022-11-21

**Authors:** Rocío Lavigne-Cerván, Marta Sánchez-Muñoz de León, Rocío Juárez-Ruiz de Mier, Marta Romero-González, Sara Gamboa-Ternero, Gemma Rodríguez-Infante, Juan F. Romero-Pérez

**Affiliations:** 1Department of Developmental and Educational Psychology, University of Malaga, 29071 Malaga, Spain; 2Department of Developmental and Educational Psychology, University of Alicante, 03690 Alicante, Spain

**Keywords:** ADHD, emotion, cognitive, theoretical models, executive system

## Abstract

Although numerous efforts have been made to deepen our understanding of the etiology of Attention Deficit Hyperactivity Disorder (ADHD), no explanation of its origins, nor of its consequences, has yet found a consensus within the scientific community. This study performs a theoretical review of various research studies and provides a reflection on the role of emotions in the origin of the disorder, at the neuroanatomical and functional level. To this end, theoretical models (single and multiple origin) and applied studies are reviewed in order to broaden the perspective on the relevance of the executive system in ADHD; it is suggested that this construct is not only composed and activated by cognitive processes and functions, but also includes elements of an emotional and motivational nature. Consequently, it is shown that ADHD is involved in social development and in a person’s ability to adapt to the environment.

## 1. Introduction

ADHD is a complex syndrome that appears in early life and has a chronic evolution [1]. Due to its impact on the day-to-day life of those who suffer from it, along with their family, friends, and colleagues, and given its widespread incidence, there have been attempts to explain and address the disorder from a range of perspectives: neuromedical and biological (including the analysis of its genetic, neuroanatomical, and biochemical bases); neuropsychological (focusing on certain altered psychological processes and executive functions, and on the facilitation of diagnosis and intervention), and psychoeducational (emphasizing aspects of psychological development and adaptation to family and school environments). At present, from an applied perspective, there is a robust body of studies that seek to detail endophenotypes in order to facilitate more precise diagnostic processes and develop new therapeutic modalities that improve the development of people in the short and long term [2]. However, most of the research on the origin of the disorder is exploratory or descriptive, principally in the form of etiological studies (which are much more abundant) and theoretical models. The latter have attempted to cover the maximum number of factors characterizing ADHD, but without the attainment of any consensus on the part of the scientific community thus far.

At the end of the 1980s, as an alternative to the attentional model, the proposal developed by Barkley [3,4] emerged. At first, it focused on people’s problems with inhibiting their responses (later understood as self-regulation failures). Self-regulation may be understood as the ability of the person to stop motor, cognitive, and emotional responses in order to substitute others that are more functional or appropriate to the specific situation that they face. During this process, the individual must carry out different operations simultaneously [5]: on the one hand, stop the predominant or immediate response; and on the other, avoid internal and/or external stimuli that may interfere in said process. During these moments of delay in response, what Barkley calls executive functions (higher and self-directed mental abilities, which help the individual to resist distraction, set new objectives or goals, and plan steps to achieve them) kick in. According to Barkley’s model, the failure of inhibition suffered by people with ADHD negatively affects the rest of the elements involved in the model. Following several revisions and the incorporation of contributions from other theoretical proposals ([6,7,8], among others), the author points to the need to promote use of the term “self-regulation”, given that the concept of behavioral inhibition can give rise to errors based on an interpretation that the relevant alterations are limited to an individual’s problem in controlling responses or behaviors, thereby diminishing attention to the cognitive, motivational, and emotional elements.

The inclusion of the term “self-regulation” in the concept of ADHD has encouraged its understanding as a disorder of psychological processes and functions that are anatomically based in the prefrontal cortex [3,4]. In recent years, consequent to the abovementioned contributions, the executive deficit has been found to take a central role in the condition [9]. This alteration causes problems in the integration and coordination of the aforementioned elements (psychological processes, emotions, and executive functions), giving rise to disorganized behavior in the natural settings wherein the person develops and must apply said elements [10]. For example: problems separating affect in interpersonal situations lead to selfish and immature social behaviors, alterations in working memory interfere with daily activities and the performance of academic tasks, and failure to internalize language results in difficulties developing self-control strategies, among others [4].

In its second version, the DSM-II [11] established hyperkinesis as the main diagnosis; the manual outlined the three categorical symptoms but excluded emotional problems. The publication of DSM-III [12] introduced a categorical approach, with a diagnosis based on the number of elements from three groups: inattention, impulsivity, and motor activity [13]. Its proposal was more specific than that of the manual’s previous version [11], an initiative that was later given support in the creation of the first symptom quantification scales by Conners [14]. The DSM-IV [15] was formulated following the application of extensive field work that made it possible to define the classification of ADHD in the three subtypes that we know today (maintained in the DSM-V) [16], on the basis of the predominance of symptoms (inattention, hyperactivity/impulsivity, and the combination of both) [13]. These recent versions of the manual relegate to the background or omit deficiencies in the emotional processing of ADHD people. This is an issue which is also reflected in a scarcity of studies seeking to define these deficiencies and to propose relevant options for rehabilitation.

In recent years, however, more and more attention has been paid to the difficulties that people show in their interpersonal relationships. During childhood, their social interaction style is frequently aggressive and/or passive, with difficulties in regulating emotions and feelings when they have to deal with interpersonal situations [17,18,19]. Their classmates often state that they are annoying and noisy, that they talk too much and interrupt conversations, that they do not respect the rules in games, and that they tend to get angry very easily, among others [20,21,22,23,24,25,26]. According to various studies, people diagnosed with this disorder tend to have low sociometric positions, despite the effort that they make to be accepted by their peers. Such difficulties persist over time and tend to increase in adolescence, becoming a prognostic factor for negative social development throughout life and increasing the risk of other associated disorders.

## 2. ADHD Conception and Emotional Control

The presence of emotional and social disturbances is hardly disputed today. However, one of the most debated issues is whether these alterations can be understood as primary or secondary deficiencies of the disorder. The first viewpoint implies that alterations in emotional recognition cannot be exclusively explained by cognitive dysfunctions, because when comparing groups diagnosed with ADHD and control groups, a similar performance is found in emotionally neutral tasks that involve cognitive skills, while performance in emotional face recognition tasks is significantly lower in the group with the disorder.

Specifically, a study by Da Fonseca, Seguier, Poinso, and Deruelle [27] indicated that children with ADHD (*n* = 27, age range: 5–15 years) exhibited a significant difference in the processing of facial expressions compared to age-matched controls (*n* = 27). In addition, it demonstrated that the ADHD cohort also had lower accuracy in identifying emotions through context, but not in identifying objects compared to healthy controls. In line with that work, Rapport, Friedman, Tzelepis, and Van Voorhis [28] indicated that adults with ADHD performed worse than controls in recognizing children’s faces. However, they found no significant differences between the two groups in the animal identification task.

These findings support the proposal that the problems in emotional recognition shown in children, adolescents, and adults with ADHD are not only the result of cognitive alterations; the deficit in emotional processing is, in fact, a primary symptom in the disorder [23,27,28,29]. Emotional recognition and emotional control are two different concepts, but closely interrelated to the extent that it is difficult to carry out emotional control and self-regulation if emotions are not adequately identified and recognized, both in the subject himself and in the people with whom he interacts.

In the second viewpoint, there is support for the hypothesis that cognitive dysfunction itself (in psychological processes such as attention, and executive functions such as self-regulation), reduces the child’s ability to recognize emotional stimuli, in addition to manifesting in their low ability to inhibit/regulate emotional responses [30]. As such, difficulties in the emotional recognition of faces and the expression and regulation of emotional states would be related to cognitive failures in the executive system (ES), which are manifested principally as attentional and self-regulation deficits. The traditional explanatory theories concerning ADHD [4,23,31] support the second viewpoint. Even those studies supporting arguments belonging to the first viewpoint do not discount the possibility that failures in emotional processing are dependent on the central deficits of the disorder; for example, there is an understanding that ADHD people are not capable of extracting sufficient information when photographs of faces are presented, owing in part to the short duration of the presentation of the experimental stimuli in the first task, which would require a large attentional load that leads to failures in recognition [27].

Thus, from either of the two viewpoints (primary/secondary), it is necessary to expand the scope of ADHD studies within a psychological and psychoeducational perspective, with the aim of increasing the attention given to emotional and social deficits; this is due to the diversity of symptoms that make it difficult for people to adapt to their environment, despite receiving pharmacological, psychoeducational, and/or multicomponent treatment [23,32]. These alterations are manifested in the ineffectiveness in the acquisition (or late acquisition) of skills such as: identifying emotional signals in one’s own body and associating them with objects or events to give them meaning, effectively recognizing emotions that others are experiencing, stopping escape responses to negative emotions, appropriately regulating one’s thinking and behavior according to the objective pursued, evaluating possible consequences of the aforesaid responses, reviewing past errors and correcting them, and/or making adaptive decisions. Alterations due to ADHD also directly influence social problems, leading to a poor ability to resolve interpersonal conflicts.

## 3. Emotions

At its inception, the proposal focused on perceptual components or physiological changes as the keys to the origin of the emotional episode [33]. Following criticism from the prevailing empirical and medical currents of the time, the author gave a central role to evaluative (cognitive) processes during the initial moments of said emotional processing. This first theoretical model constituted the basis on which most subsequent proposals have been constructed, which have varied between physiological activation and cognitive assessment of the emotional situation, including studies on the role of evolutionary pressure, the brain areas involved, the role of the body as the encoder of physiological signals, or the importance of facial feedback (for an in-depth chronological analysis of the various approaches related to this construct, see Roselló and Revert [34]). In recent decades, the dedication of researchers and academics to the study of the relationships between emotion and cognition has highlighted the ubiquitous role of emotion in human cognition and behavior, whether at the perceptual and attentional levels [35,36], in relation to memory [37], or in reasoning and decision-making [38,39]. At the neural level, it seems increasingly clear that emotional and cognitive processes cannot be separated, given the interaction between the neural bases of each, which situates them as non-modular elements [40,41,42], or as dimensions of a whole.

With regard to the proposals of current research, these integrate elements of an emotional, cognitive, and motivational nature in their answers to the why and how of emotional processing. The core-affect theory [43] assumes the existence of: (a) an affective or neurophysiological state that occurs as a result of the combination of hedonic values (pleasure–displeasure) and arousal (activation–relaxation), that is experienced in the body and that precedes and interacts with other components, these being: (b) attribution to the stimulus that gives rise to it (the core-affect is perceived as a response to a trigger, object, or event that arouses it and that possesses an implicit affective quality); (c) appraisal (the subjective assessment that the person makes of said stimulus); (d) tendency to action (predisposition to respond based on said subjective assessment); (e) somatic, facial, and/or vocal changes (physiological manifestations of the emotional state that the person experiences); and (f) subjective experience (also understood as emotional meta-experience, or categorization/attribution of meaning to the emotional episode that results from the prior elements). The first two elements (core-affect and appraisal) trigger all emotional processing [41]. The first element is nonspecific and internal (identified with sensations or physiological changes experienced by the body) and can be attributed to a cause (an external or internal stimulus that provokes it). The second (the affective quality that we grant after experiencing these sensations) is external and inherent to the stimulus, but it can influence the first by lending it meaning in future similar situations.

Along similar lines, corporeal theories—specifically, the somatic marker hypothesis [44]—try to clarify the role of emotions and feelings in decision-making. Damasio considers that any human decision is the result of emotional and cognitive mechanisms. The ventromedial prefrontal cortex is responsible for establishing the link between a neural map for a specific situation and the emotion that was associated with it in the past [45]. When evoking the specific situation, emotional images are recovered from sensory cortices (that is, the somatic pattern corresponding to the emotional episode is reactivated), giving the person signposts (or markers) for processing the information and generating an appropriate response (choose one or the other option). Said marker (the core-affect, in Russell’s model) serves as an alarm to anticipate the scenario in which the person must function (providing information about the appraisal or affective quality of the stimulus that triggers it), preparing the way for the activation of cognitive processes and functions; these will restrict response options after carrying out an evaluation of the emotional situation in which they are immersed (analyzing the stimulus or stimuli that trigger(s) it, the body changes experienced, and the situations experienced in a similar way in the past; assessing possible responses and their consequences—the benefits and the losses, as well as anticipating the behavior and feelings of the other people involved; taking into account possible errors), until they arrive at the most adaptive response. Somatic markers are generated from two types of events: primary inducers (stimuli that have been associated—innately or through learning—with pleasant or aversive states, and that generate automatic emotional responses) and secondary inducers (generated from the personal or hypothetical memory of an emotional event) [46,47].

In relation to markers, of particular interest are the theses of Kaboodvand, Iravani and Fransson [48] on functional connectivity (FC) representing the level of synchronization between different brain regions/networks; measures of brain FC can serve as a useful tool for diagnosing and predicting the course of the disease, and are useful for developing individualized therapies. In this same direction of identifying brain biomarkers using functional magnetic resonance imaging techniques is the research by Uddin, Dajani, Voorhies, Bednarz and Kana [49]. Identification of brain biomarkers for ASD and ADHD could aid in the objective diagnosis, monitoring of treatment response, and prediction of outcomes for children with these neurodevelopmental disorders. However, at present, the field has yet to identify reliable and reproducible biomarkers for these disorders and must address issues related to clinical heterogeneity, methodological standardization, and validation before further progress can be made.

The differences that exist between primary and secondary emotions and/or feelings, and of which emotions/feelings belong in one or the other group, has likely been among those most discussed in recent years. In 1969, Ekman [50] proposed a classification of six innate and/or basic emotions (enjoyment, sadness, anger, fear, surprise, and disgust), which he then went on to review and expand. In 1982, he pointed out that emotions have perceptual, physiological, expressive, cognitive, and subjective components, which complicates the task of differentiating emotion and feeling [51]. These emotions, considered as basic because they are universal, cross-cultural, automatic, and innate (aimed at ensuring the survival of the species), are increasingly subject to cultural elements [52] and perform functions that go beyond species survival (communicative, social, and motivational). Barret [53] considers that what is innate is in fact a state of sensitivity or excitability to react to what is pleasant/unpleasant, and not so much the emotion itself. Therefore, it is not possible to create a classification of emotions on the basis of universality and survival, taking as given that the rest of the emotions are subject to social, cultural, learned, and conscious elements (those referred to in the literature as secondary or self-conscious emotions). As current theories suggest [44,45], it is necessary to address the classification of emotions and feelings (we will ignore the primary and secondary terms, given the possibility of provoking semantic errors) as a continuous process. Emotions must be understood as rapid and corporal affectations, which occur even within newborns [51,54]. Incrementally, throughout development, emotions (according to current theories) are nourished by experience and cognition, becoming more and more elaborate, as well as dependent on context, thought, memory, and moral schemes, giving rise to what we understand as feelings.

The relationship between emotions, feelings, and cognition/memory is a broad one. Emotion and feeling affect emotional evaluation and information processing [55]; in other words, the greater the emotional assessment of the event, the more intensely it is remembered. That is, the affective load of the event can render its memory more lasting [56]. Among the various cognitive processes, or, more specifically, among executive ones, decision-making may be the most studied function with regards to the influence of emotional processing. When the person is faced with the need to pursue one option or another, the orbitofrontal cortex restrains the predominant impulse or response and carries out a process of neurobiological integration of the reward systems, restraining or activating the amygdala and the nucleus accumbens—emotional and corporeal elements. Research has observed that the nucleus accumbens is activated by expected rewards or benefits, and the amygdala by costs or losses, as well as feared stimuli [57]. In addition, the anterior cingulum evaluates the costs and benefits of responses, and thus resolves the conflict or determines the best decision [51]—cognitive and motivational elements. The restraint affected by the orbitofrontal cortex (which gives rise to the capacity for inhibition or self-control) allows responses to be delayed in order to analyze the different possibilities by comparing the rewards, until one of the last two options falls below the minimum reward limit [58]. The dorsolateral prefrontal cortex supports said assessment process by means of keeping information online in order to enable its manipulation (working memory or WM), thereby facilitating emotional regulation [59]. As may be observed in the above explanation, decision-making is a complex process, influenced by elements of various kinds: bodily, cognitive, and emotional (as well as contextual, cultural, and moral).

Analysis carried out on the processing of emotions and feelings in people reveals the importance of ES processes and functions in the adequate expression and regulation of responses, as well as in appropriate assessment, planning, and decision-making, especially in social situations and emotionally charged ones, due to their high contextual dependence and the number of brain processes that must work concurrently. The conception of emotions as body signals oriented toward action or survival lacks richness in this contemporary research context. Affective experiences involve the activation and coordination of neural systems and internal and external information processing, which have to work in a coordinated and networked manner, with the boundaries between one thing and the other becoming blurred. This analysis becomes complicated when we consider how the brain of a person who presents a neuropsychological alteration will behave—specifically, during the first years of vital development, when the brain is especially susceptible to external influences.

## 4. Proposal for an Integrative Cognitive-Emotional

In accordance with the above analysis, this study has assumed the following objectives: first, to review various theoretical models that contribute a cognitive and/or motivational perspective to the hypothesis of the disorder’s origins, in addition to recent research offering an emotional focus; second, to outline an updated explanation of the concept of ADHD, starting from a psycho-educational perspective that takes account of emotional processing as one of the affected elements within the executive system of someone suffering from the disorder; third, to promote new lines of research that supply adequate tools for the diagnosis and treatment of ADHD to the professionals working with the disorder.

The emergence of the cognitive-emotional proposal in the conceptualization of ADHD (see Figure 1) can be seen in the cognitive models elaborated by Barkley [60], in his hybrid model of self-regulation and executive functions, and by Lavigne and Romero [61], in their model of cognitive deficit in the executive system; it may also be traced back to the motivational and behavioral contributions made by Sonuga-Barke [62], in his model of unique deficit in delay aversion, and the later revision of the same, with which he created a dual model. Additionally, we refer back to certain studies (behavioral, in general) supporting the hypothesis of alteration in emotional expression and regulation within ADHD; specifically, we focus on the integrative theory of ADHD by Nigg and Casey [63] and on Thomas Brown’s model [31], dealing with activation, concentration, effort, emotion, memory, and action in ADHD. In parallel with this, we analyze models or theoretical proposals that address the emotional processing of normotypical individuals from a dimensional perspective: we start from the corporeal theory of somatic markers by Damasio [44], as it is the most widely accepted contribution in the field, and we also review Russell’s core-affect model [43,64].

According to Lavigne and Romero’s model [61], the executive system is framed within information processing psychology as the system involved in the controlled processing of actions aimed at achieving a goal. This system intervenes when the response is aimed at achieving a goal that is not routine (hence, it is not a habit, nor is it found in the person’s behavioral repertoire) and that requires planning; there is a time delay between the elements of the behavioral sequence (stimulus, response, and consequences), giving rise to a conflict between the immediate and the long-term consequences [65,66]. There is consensus in the view that the ES is in charge of planning complex responses to achieve objectives, programming the necessary actions to carry them out, monitoring their implementation, controlling the interference of irrelevant stimuli and correcting errors, and incorporating new responses based on the demands of the environment [66]. The complexity that characterizes this construct, given the multiple cognitive, emotional, and motivational capacities implied, has resulted in efforts to separate out its components and functions [67]. Regarding those ES components that are relevant to ADHD, and with consideration to the idea that the brain functions as an organized and systemic network, we consider that it is necessary to differentiate the following:(1)Structural/anatomical components of the mind: areas of the frontal lobe and their connections with the basal ganglia, limbic system, and cerebellum.(2)Components of consciousness (psychological and motivational processes): perception, attention, working memory, internalization of language, theory of mind, processes of analysis and synthesis, motivation.(3)Emotions: joy, sadness, anger, fear.(4)Feelings/affections.(5)Executive functions: self-control and self-regulation, planning/organization, flexibility, decision-making, metacognition.

The ES is made up of anatomical structures (1) that are innervated by neurotransmission pathways (dopamine and norepinephrine, principally) whose functions are materialized in the coordination of psychological processes, or in components pertaining to consciousness (2) and emotions/feelings (3 and 4). The combination of these cognitive-emotional elements activates (executive) functions (5) that promote the appearance of skills, strategies, and therefore adaptive behaviors in the person, generating, supervising, regulating, executing, and readjusting the response [68] in a great variety of tasks, dependent on contextual demands. The neurochemical alteration that characterizes ADHD hinders the overall work of the system in the context of both the poor quality of the interconnections between cortical and subcortical structures and the limited integrative role of the prefrontal cortex. Such alterations translate into regulatory problems in the processing and emission of responses (cognitive, motivational, emotional, bodily, and behavioral), the defining features of which result in the symptoms that characterize ADHD: failures of inhibition capacity, low impulse control, difficulties in sustaining attention, lack of verbal control, motor restlessness, emotional lability/reactance and/or problems with emotional and behavioral control, and difficulties related to learning, among others.

The integrated conception of the ES as a processing system must presuppose the parallel work of all the elements that comprise it. However, likely due to the preeminence of the cognitive paradigm in recent decades, research contributions have examined to a greater extent those aspects related to failures in reasoning, learning, or behavioral adaptation among people with ADHD, particularly in the early stages of development. However, the elevated presence of comorbid disorders throughout life, in addition to the chronicity of the symptoms themselves (even in spite of multicomponent and contextualized treatment), reveal the need to further refine the theoretical basis underpinning diagnostic and procedural processes. As such, we consider it necessary to shift the attention of academics, clinicians, and researchers from the cognitive and behavioral elements that have traditionally received in-depth study and that have even determined the acronyms by which the disorder is known. Instead, we propose interpreting the disorder’s genesis and development through an examination of the failures experienced by the ES in regulating the overall work of its elements. Accordingly, we will start from the deficit in self-control and self-regulation (cognitive, behavioral, and emotional) in establishing the guidelines to this approach (without diminishing the rest of the elements integrated into the model or losing sight of the overall work that the system must perform).

In line with Barkley’s [69] proposal, and on the basis of the synthesis established therein, self-control may be understood as an executive function (EF, hereafter) directed at oneself and aimed at changing subsequent behavior in order to obtain the maximum benefit from positive results, in the short and long term, based on self-regulation strategies and requiring motivational processes that support the identification and maintenance of long-term rather than immediate rewards. This necessitates sufficient inhibitory capacity to control the time between an event and the person’s response (latency period), in addition to the capacities of retrospection (analysis of similar events in the past) and anticipation (of future consequences). These capacities cannot function adequately if the psychological processes on which they depend (WM, for example) are altered. Such an alteration also limits the ability of the person to integrate and imitate functional responses, to formulate rules or action plans, or to apply moral logic (among other effects), leading to an increase in learning situations that are dependent on trial and error, and increasing the probability that people make the same mistakes again and again. Specifically, self-control and emotional self-regulation can play a very significant motivating role in the accomplishment (or not) of socially acceptable specific responses (such as guiding one’s excitement or maintaining perseverance in situations of delayed reward).

In natural settings, the restraint or regulation of the response (understood in behavioral terms) allows the activation and coordination of the components of consciousness in order to carry out the pertinent mental operations (integration, evaluation, information processing, and generation of adaptive responses). Specifically, in situations with affective load (generally, social situations and those in which conflicts appear), the task of the ES expands due to the number of variables involved. As indicated in revised theoretical models [44,64], cognitive and emotional elements (as well as motivational and corporeal elements) overlap for the benefit of the person comprehending the situation experienced, identifying the response that will entail greater benefits. In the case of ADHD, executive failures, in addition to alterations in the person’s motivation to withstand delay in rewards, increase the risk that the ES could lack sufficient time or resources to carry out its function in a coordinated manner, hence the unpredictable and maladaptive responses that are observed in diagnosed people (appearing, especially during childhood and adolescence, as immature, transgressive, and even aggressive), thereby incurring the characteristic problems of the disorder in the social sphere, even when they are carrying out the conventional treatment programs on a systematic and prolonged basis.

## 5. Prospects for Future Research

It is necessary to make clear that the proposed conception is based on the analysis of a single attention deficit and hyperactivity disorder (understood in the literature and diagnostic manuals as a combined subtype); therefore, in future research, it would be pertinent to review the above proposals and to clarify the differences among the relevant subtypes or differentiated disorders (Barkley [30]; Capdevila [71]; Tirapu [72]). Regarding subtype differentiation in the diagnosis of ADHD, it is worth bearing in mind the research by Iravani, Arshamian, Fransson, and Kaboodvand [73] on the importance of functional magnetic resonance imaging techniques for the differentiation of ADHD subtypes. The results obtained allow us to be optimistic, as they demonstrate the potential of this new modeling framework to unveil hidden neurophysiological profiles and to establish tailored clinical interventions.

## 6. Conclusions

In view of the above, it is necessary to revise the parameters of diagnosis and intervention to include instruments that specifically analyze emotional and social variables within the assessment protocols. With regards to the combined treatment options (pharmacological, psychoeducational, and context-oriented), we advise the following: once the categorical variables of the disorder have been rehabilitated (stimulation of psychological processes and executive functions), guidelines for behavior modification and behavioral management have been established, and a psychoeducation process has been carried out in relation to the understanding of ADHD and the development of strategies to improve learning, autonomy, and adaptation to the person’s different environments, it is necessary to include objectives aimed at rehabilitating the processing (identification) and regulation (expression and management) of primary emotions and feelings (especially when they are negative), in addition to socio-cognitive (theory of mind, empathy, pragmatics) and communicative (verbal and non-verbal conversational skills) elements to support adequate conflict management, through the application of cognitive-behavioral techniques and therapies that must be put into practice individually, as well as within the peer group.

With regard to intervention programming, it will be necessary to adjust the objectives according to the stage of evolution of the person’s disorder, the impact that the symptoms are having, and the degree of knowledge concerning the disorder possessed by agents involved in the person’s education (family and school, mainly). The first phases of treatment should be aimed at offering an initial explanation of the symptoms and the therapeutic strategies to be used (for the purpose of providing those involved with sufficient knowledge to establish the parameters of the problem/objectives, which will be agreed upon over the course of treatment). Subsequent to this, work with the child/adolescent will be individual (behavioral and cognitive variables), with a twofold objective: on the one hand, to rehabilitate categorical variables and provide the person with strategies to promote their autonomy in different contexts; on the other hand, to grant adults the necessary skills to evaluate, guide, and modify the child’s/adolescent’s behavior and performance.

As mentioned above, the work of the ES increases in social situations, owing to the number of variables involved and the immediacy and unpredictability with which events occur in these types of scenarios. Therefore, without losing sight of those elements addressed thus far, it will be necessary to provide a course of emotional treatment (conducted individually, at first) in addition to treatment of the socio-cognitive and socio-communicative type (in a small group and always led by the therapist), preferably on a preventive basis. Throughout primary schooling (even in early years), it is possible to conduct an educational process targeting the processing of emotions that promotes not only the relevant skills, but also the exchange of information among family members, thus supporting the early detection of social problems and the prevention of other pathologies that may develop during adolescence and adulthood.

In summary, the analysis of ADHD from a systemic perspective—in which it is understood as an alteration that has its origin in the ineffective coordination of component elements of the executive system (pertaining to variables of a cognitive, emotional, and behavioral nature)—supports the development of plural considerations in the understanding, diagnosis, and treatment of the disorder, thereby increasing the efficiency and, especially, the effectiveness of the interventions, in addition to reducing possible adverse effects in the future.

## Figures and Tables

**Figure 1 ijerph-19-15421-f001:**
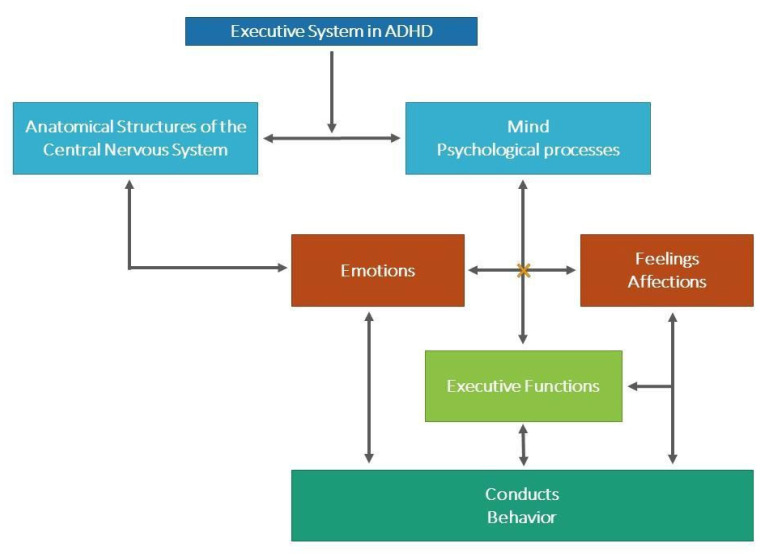
Proposal for an integrative cognitive-emotional conception of ADHD. ES: planning complex responses, programming the necessary actions to carry them out, monitoring their implementation, controlling the interference of irrelevant stimuli, and correcting errors or incorporating new responses according to the demands of the environment [70]. CNS: areas of the Frontal Lobe and its connections with the Basal Ganglia, Limbic System, and Cerebellum. Psychological Processes: Perception, Attention, Working Memory, Language Internalization, Theory of Mind, Analysis and Synthesis Processes, Motivation. Emotions: Joy, Sadness, Anger, Fear. Feelings/Affects: Thought Emotions. EF: Self-Control and Self-Regulation, Planning/Organization, Flexibility, Decision-Making, Metacognition. Behaviors. The bidirectional arrows are intended to signify the reciprocal influences that occur between the different structures, psychological processes, and behaviors involved, to the extent that one or the other takes place. ×: alteration.

## Data Availability

Not applicable.

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
