# Peer review of "Proposal for an Integrative Cognitive-Emotional Conception of ADHD"

_ijerph, 2022, doi:10.3390/ijerph192215421_

Round 1
Reviewer 1 Report
It is a great pleasure to be invited to read this interesting paper describing the proposed model.
1. Please do not use the word subject(s) when discussing the young people, it is a de-humanising term that psychologists and other health professionals are advised by their professional organisations to avoid. The word participant(s) is appropriate for volunteers taking part in a study, or patient(s) if the data is collected via their health professionals.
2. The conclusion should be extended or amended to make it clear what the new information or theory is: "Consequently, it is shown that ADHD is involved in the social development of the subjects and in their ability to adapt to the environment." Nb: Opening line by Humphries et al 2016: "Attention-deficit/hyperactivity disorder (ADHD) reliably predicts social dysfunction, ranging from poor social competence and elevated peer rejection to inadequate social skills." Humphreys, K. L., Galán, C. A., Tottenham, N., & Lee, S. S. (2016). Impaired social decision-making mediates the association between ADHD and social problems. Journal of Abnormal Child Psychology, 44(5), 1023-1032.
3. Line 34, Replace the word evolution with the word development.
4. Line 101, Remove the "etc" from the list of examples. There are other sentences with this issue, please check through the whole manuscript.
5. Line 102, Remove the word materialized and replace it with the word resulted: "..into the disorder’s etiology materialized in.." When your sentence clearly states that these are examples, the read understands it is not the full list, so you do not have to add etc.
6. Lines 102 - 123, Summarise the long history of development to get to your point on lines 124 - 127 quicker.
7. Lines 128 - 132, Use a native English speaker who understands this area of research or a professional editing service to re-phrase this sentence: "As may be observed in the approach of models explaining the etiology of ADHD, the interest of most researchers has focused, firstly, on the triad of core symptoms (inattention, hyperactivity, impulsivity), with the assumption that other altered elements would constitute collateral symptoms of the aforesaid deficits and, therefore, with an orientation on cognitive and behavioural causes."
8. Line 138, Remove "etc" It is legitimate to give some examples and let the reader find more in the cited papers.
9. Line 134, The phrase "predominantly aggressive" would be appropriate if you are describing behaviour under duress, but this sentences makes it appear you are suggesting the young people are aggressive every day and most of each day. This is too much. It might be better to express the daily social behaviour as intrusive and / or passive.
10. Lines 157, 173, 335, Remove the word healthy and replace it with neurotypical.
11. There are small grammatical errors, for example line 291, "...most commonly discussed recent years..." should have the word in inserted between discussed and recent. Please review the whole text for similar issues. Another example: Line 307, delete in after within.
12. The lengthy histories of each topic are interesting if the main aim of this paper is to provide the history, but that was not the stated purpose, so it would be helpful to summarise the early development and focus on the most recent findings.
13. Check abbreviations, for example the abbreviation ES is introduced twice, first on line 194 then again on line 374.
14. Lines 404 - 430, Is all this information from one source? This does not present a robust synthesis of multiple sources. Perhaps more supportive citation could be found? The same comment applies to lines 436 - 453.
15. Lines 456-463, This sentence is too long, so the who does not apply to the original subject of the sentence. Please start a new sentence for this part: "..who, thereby, incur the characteristic problems of the disorder in the social sphere even when they are carrying out the conventional treatment programs on a systematic and prolonged basis."
16. The conclusions section is too long. Put any new points into the discussion with appropriate citation, then present the summary of the key points as the conclusion. Do not introduce new information in the conclusion. Do not add citation to the conclusion, this should be the new ideas and all supportive citation will already have been given in the discussion.
Fig 1. Revise the image, some of the text has been distorted.
Reviewer 2 Report
Comments to the Author
Summary
The author performed a systematic review and provided a theoretical reflection on the role of emotions in the origin of the ADHD. To broaden the perspective on the executive system in ADHD, the author constructed a novel theoretical model which includes the elements of emotional and motivational nature. The author suggested that it is necessary to analyze emotional and social variables within the ADHD assessment protocols systematically. Furthermore, the author concluded that ADHD could be understood as an alteration, which originates from the ineffective coordination of component elements of the executive system.
However, I have major concerns about the structure and logic of the entire paper, as its rather confusing. Major comments are presented here.
Major concerns
1. The model for integrative cognitive-emotional conception of ADHD proposed by the authors is not clearly explained and illustrated. For example, the authors did not indicate what the symbols and arrows within the model diagrams means. Also, the text in the diagram is not friendly presented (some words are even blurred or not in the box). In this article, the author used a long length of illustration (nine paragraphs) for “introduction” section. However, the model that requires detailed illustration is not explained fully.
2. What does the role of the “emotion” section play in the whole article? This section provides a popularization of science about the development, classification of emotions, as well as the importance of Executive System processes and functions in the regulation of emotional expression and emotional response. Above all, this section looks like an independent part and not significantly connected with other parts of the manuscript.
3. The rationale of putting this article in the issue of “Emotion Regulation in Children and Adolescents” is not clear, as “emotional regulation” or “regulating the emotion response” were mentioned for only few times in this manuscript.
4. The superior-subordinate relationship of some concepts is not clear. For instance, the relationship of psychological process, executive function, and executive system.
5. The logic and structure of the whole manuscript is not clear enough, and the subheadings and sub-points are ambiguous and obscure.
6. There are too many long and complicated sentences in this manuscript which lead to a low readability.
Reviewer 3 Report
The authors provide a review on the literature of the attention deficit hyperactivity disorder (ADHD) to strive to update the explanation for ADHD concept and promote new line of research. In the following I provide my comments on this manuscript.
Major
1. I recommend the authors to condense the manuscript and keep only the parts that are absolutely necessary. At its current form it is more suitable for a book chapter rather than an article.
2. The review literature can be improved by mentioning the conclusion of the past research rather than reporting their stats. This aspect of the manuscript has to be amended.
3. Authors did not review enough the ADHD neuroimaging literature. Functional neuroimaging deserves a better review here. Please see the following articles:
· Kaboodvand N, Iravani B, Fransson P. Dynamic synergetic configurations of resting-state networks in ADHD. Neuroimage. 2020;207:116347.
· Uddin LQ, Dajani DR, Voorhies W, Bednarz H, Kana RK. Progress and roadblocks in the search for brain-based biomarkers of autism and attention-deficit/hyperactivity disorder. Transl Psychiatry. 2017 Aug 22;7(8):e1218.
· Uddin LQ, Kelly AMC, Biswal BB, Margulies DS, Shehzad Z, Shaw D, et al. Network homogeneity reveals decreased integrity of default-mode network in ADHD. J Neurosci Methods. 2008 Mar 30;169(1):249–54.
4. How about the heterogeneity in the ADHD cohort needs to be reviewed
· Iravani B, Arshamian A, Fransson P, Kaboodvand N. Whole-brain modelling of resting state fMRI differentiates ADHD subtypes and facilitates stratified neuro-stimulation therapy. Neuroimage. 2021 May 1;231:117844.
Minor
1. Please make sure that the texts in the Figure1 fits within the boxes. Some letters of the word “the” is now extruded from the box. Or the letters in the first box are overlapping.
a. What are the colors of the boxes representing here. If they mean something they should be mentioned in the figure legend. Or if now use same color for all boxes.
b. You mentioned structure of the CNS, how about the function of CNS, they are not same thing. They are to a large extend overlapping but not entirely.
At the current form of the manuscript, I can’t recommend this work for publication.
Round 2
Reviewer 1 Report
The authors have addressed some of the points effectively, but not understood some of the suggestions (points 3, 4, 8). Other advice they have responded that they choose to ignore, for example (point 16), that the conclusion should not include new information and not include citation which is based on standard practice in good scientific writing.
It would benefit from more work.
Best wishes
Author Response
Consulte el archivo adjunto

Reviewer 2 Report
Major concerns:
(1) In the Figure 1 legend, the author has added the explanations for each component of ES, but we think it is also necessary to add explanations for some symbols such as the two-way arrows and the yellow cross in the model diagrams.
(2) In the second section ‘ADHD conception and emotional control’, the author spent too much space describing two empirical researches on emotion recognition in ADHD patients.
(3) The title of Section 2 is talking about emotional control, however, those two cited studies are focusing more about emotional recognition (a kind of emotional processing) problem in ADHD. However the "emotional recognition" and "emotional control" are two different concepts. The cited researches cannot fully support the topic of emotional control in ADHD.

Reviewer 3 Report
Authors have improved the manuscript considerably, however there are some points that need to be addressed, before I can recommend this manuscript for publication.
1. I understand that proposing a new concept needs an extensive argument, however, I do believe that these arguments can be written in more concise manner. So, a more effort on that front would be appreciated a lot.
2. Page 2, line 91: The sentence “The inclusion of the term "self-regulation" in the concept of ADHD has encouraged its understanding as a disorder of psychological processes and functions that are anatomically based in the frontal lobe.” lacks reference. Given that the sentence here is discussing the “self-regulation” and not the other aspects of ADHD, using the term “prefrontal cortex” sounds more suitable rather than general term of “frontal lobe”.
3. Language editing is required. For example, line 136, there is misspelling of the word “frequently”.
4. In the first round of revision, I suggested that the authors to summarize that past literature findings rather than reporting the detailed tasks and stats. There is an improvement in this matter, yet it is not enough. For example, I suggest page 4 lines 154- 174 to be summarized as follows:
Specially, Da Fonseca and colleagues [27] indicated that ADHD children (n = 27, age range: 5-15 years) exhibited a significant difference in the processing of facial expressions when compared to age-matched controls (n = 27). Moreover, Da Fonseca and colleagues further demonstrated that ADHD cohort had also lower precision in the identification of emotions through the context but not in the identification of objects compared to healthy controls.
I recommend summarizing all the past findings in this manner. This will help to save space and be clearer rather than cluttering the manuscript with stats and detailed task design. If readers are interested in the details, they may always refer to the original paper.
